# Downscaling of Regional Air Quality Model Using Gaussian Plume Model and Random Forest Regression

**Marcin Kawka** [1,*] , **Joanna Struzewska** [1,2] **and Jacek W. Kaminski** [1]

1 Institute of Environment Protection—National Research Institute, Kolektorska 4, 01-692 Warsaw, Poland
2 Faculty of Building Services, Warsaw University of Technology, Nowowiejska 20, 00-653 Warsaw, Poland
* Correspondence: marcin.kawka@ios.edu.pl

**Abstract:** High $PM_{10}$ concentrations are still a significant problem in many parts of the world. In many countries, including Poland, 50 µg/m³ is the permissible threshold for a daily average $PM_{10}$ concentration. The number of people affected by this threshold's exceedance is challenging to estimate and requires high-resolution concentration maps. This paper presents an application of random forests for downscaling regional model air quality results. As policymakers and other end users are eager to receive detailed-resolution $PM_{10}$ concentration maps, we propose a technique that utilizes the results of a regional CTM (GEM-AQ, with 2.5 km resolution) and a local Gaussian plume model. As a result, we receive a detailed, 250 m resolution $PM_{10}$ distribution, which represents the complex emission pattern in a foothill area in southern Poland. The random forest results are highly consistent with the GEM-AQ and observed concentrations. We also discuss different strategies of training random forest on data using additional features and selecting target variables.

**Keywords:** random forest; Gaussian plume; GEM-AQ; downscaling; $PM_{10}$

## 1. Introduction

$PM_{10}$ refers to particulate matter with a diameter of 10 micrometres or less [1]. These particles come in many sizes and shapes and can comprise hundreds of chemicals. Many of these chemicals are emitted directly from an anthropogenic source, such as construction sites, unpaved roads, fields, smokestacks, or fires. $PM_{10}$ is a major air pollutant that can significantly impact human health, including respiratory and cardiovascular diseases [2]. Within Europe, Poland and, in particular, its southern provinces are in focus due to their high industrialization [3], and a large number of the population is affected by high $PM_{10}$ and $PM_{2.5}$ concentrations [4]. Therefore, accurate and detailed information on $PM_{10}$ concentrations is essential for assessing and managing the impacts of air pollution.

In recent years, advances in remote sensing and satellite data have enabled high-resolution $PM_{10}$ concentration maps at regional and global scales [5,6]. These maps can provide valuable information to policymakers, researchers, and the general public. They help to plan and evaluate efforts to improve air quality and protect human health. However, due to inherent remote sensing limitations, they remain a temporal snapshot (or a mosaic of snapshots), sometimes misinterpreted as monthly or annual average concentrations. Still, additional means are needed to meet the demand for high-resolution concentration maps [7].

Using mathematical equations, air quality models can simulate the transport and dispersion of $PM_{10}$ particles in the atmosphere. These models estimate $PM_{10}$ concentrations at high spatial and temporal resolutions using meteorological data, emission inventories, and other inputs. This method can generate $PM_{10}$ concentration maps from regional to global scales. While air quality models can be accurate, they require significant computational resources and depend on the quality of input data. There are several classes of air quality models [8], including CTM (chemical transport models), which are based on mass and

momentum conservation equations and are the most universal and general. Since pollutant transport is highly linked to meteorological conditions, CTMs are often run together with meteorological models (online approach) or use the results of a meteorological model (offline approach).

Gaussian plume models are analytical and widely used due to their low computational demands [9]. These models provide an analytical solution to pollutant transport equations, assuming one of the atmosphere stability classes. They are usually applied for a single stack to assess its environmental impact [10].

Another rapidly emerging approach for obtaining air quality concentration maps is the application of machine learning techniques (also known as data-driven modelling) [11]. In this case, the user is only responsible for defining the input data (such as meteorological parameters, land use, and emission inventory data) and target variables (usually concentrations). These models act as black-box models and are trained using supervised learning processes. Depending on the output type, they can be classified as classification or regression models. The most popular model types include neural networks (focusing on LSTM networks [12]), random forests [13], and spatial kriging algorithms [14,15].

Data fusion methods can combine data from different sources [16], such as ground-based monitoring, satellite remote sensing, and air quality models, to generate high-resolution $PM_{10}$ concentration maps. These methods use statistical and machine learning techniques to merge the data and estimate $PM_{10}$ concentrations at locations where data are missing or incomplete. While data fusion methods can improve the accuracy and spatial resolution of $PM_{10}$ concentration maps, benefiting from multiple approaches, they must be applied cautiously as they often produce non-physical results [17,18].

The objective of this paper is to demonstrate an application of the random forest as a data fusion technique for results originating from two sources (the regional GEM-AQ model [19] and the local Gaussian plume model). Section 2.2 describes the details of the GEM-AQ model, while Section 2.3 describes the Gaussian model. In Section 2.6, we discuss several approaches to the application of random forests. The proposed approach was tested on a regional domain located in a diverse area of southern Poland, routinely modelled with a coarse-resolution regional model. The proposed approach can also act as a form of downscaling for air quality model results. This region was chosen due to data availability and reported air-quality issues in the past. Observation and emission data from the whole year of 2021 were used.

## 2. Data and Methods

This study used the GEM-AQ model 24 h forecast from the operational run and observations from the national air quality monitoring network. The study period covered the year 2021.

### 2.1. Study Area

The study area is located in southern Poland, covering an area of around 5300 km$^2$ in the Silesian and Lesser Poland provinces (Figure 1). The area is populated with almost 1.7 M people. The largest cities include Bielsko Biała (170,000 inhabitants), Rybnik (140,000 inhabitants) and Jastrzębie-Zdrój (91,000 inhabitants). The northern part of the study area covers the upper Vistula Valley, which has a high urbanization level (Figure 2). In contrast, the southern part reaches the Carpathian mountains, which limit air mass exchange (Figure 3). Temperature inversion is frequently observed, especially in foothill valleys in the winter period. This fact limits boundary layer mixing and contributes to poor air quality [20]. As a consequence, cities within the study area suffer from poor air quality due to high $PM_{10}$ concentrations [4].

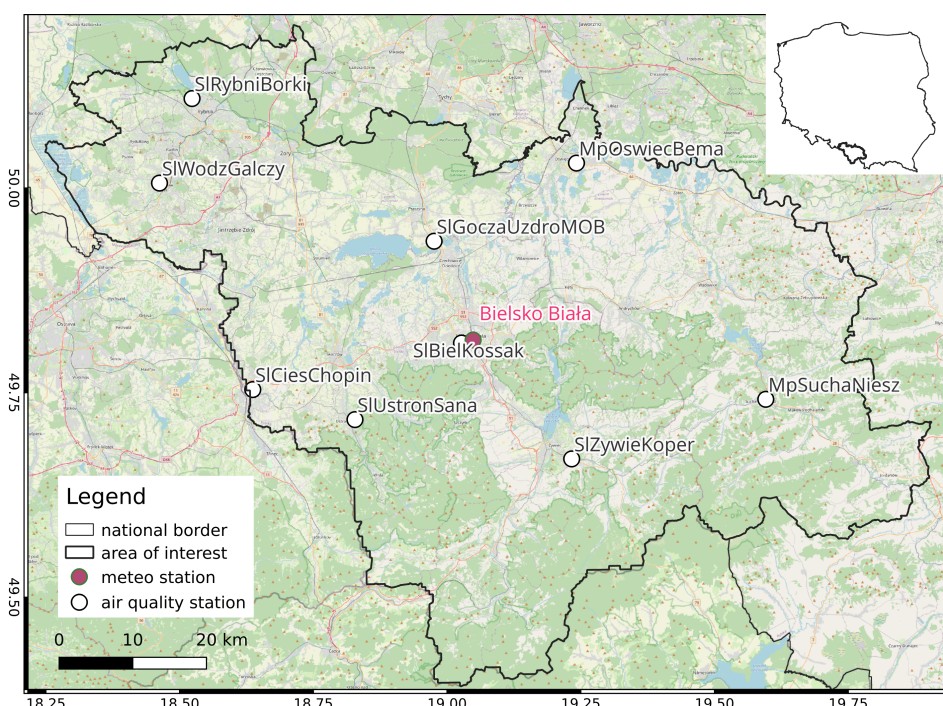

**Figure 1.** Location of the study area with meteorological and air quality stations.

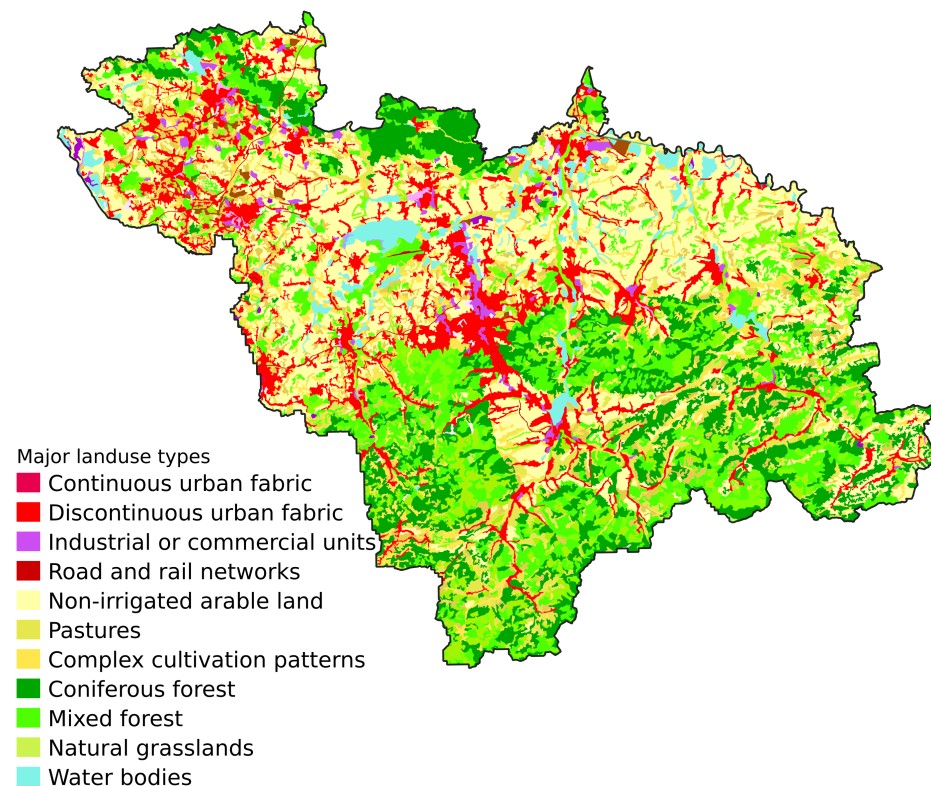

**Figure 2.** Landuse classes within the study area.

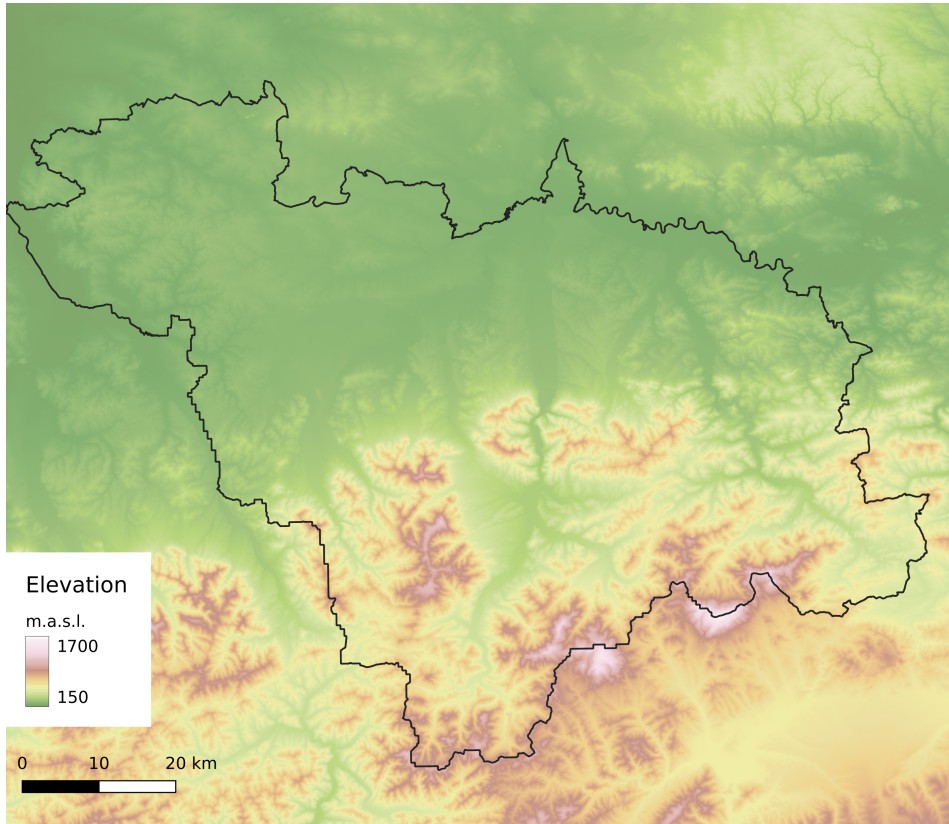

**Figure 3.** Land elevation of the study area.

### 2.2. The GEM-AQ Model

The GEM-AQ model is a semi-Lagrangian chemical weather model in which air quality processes (chemistry and aerosols) and tropospheric chemistry are implemented online in an operational weather prediction model, the Global Environmental Multiscale (GEM) [21] model, which was developed at Environment Canada. The gas-phase chemistry mechanism used in the GEM-AQ model is based on a modified version of the Acid Deposition and Oxidants Model (ADOM) [22], where additional reactions in the free troposphere were included [19].

The GEM-AQ model is set up to perform calculations using 28 vertical layers, out of which the lower 21 layers are classified as the troposphere.

Emission data from the Polish national emission inventory drive emission sources within the model. These data are based on annual reporting obligations the facilities' owners fulfil. Annual emissions are transformed into monthly emission rates using the weighting factor from annual emission profiles. Emission profiles are assigned to the so-called SNAP emission categories [23,24]. Within this work, SNAP 01 (combustion in the production and transformation of energy), SNAPS 02,03,04 (industrial and non-industrial combustion plants) and SNAP 07 (road transport) were included. Resulting $PM_{10}$ time series from GEM-AQ model are presented together with observed ones in the Appendix A.2.

### 2.3. The Gaussian Plume Model

A Gaussian plume model is a widely used mathematical model for predicting the dispersion of pollutants in the atmosphere. The model assumes that the dispersion of pollutants can be approximated as a two-dimensional Gaussian distribution, which spreads out in a pattern similar to the shape of a bell curve.

The Gaussian plume model is based on the idea that a combination of atmospheric turbulence and wind patterns determine the dispersion of pollutants. The model considers source strength and height, wind speed and direction, and atmospheric stability. The topography of the modelled area is not considered.

The Gaussian plume model is used in various applications, including air quality assessments for industrial facilities [25], roadway emissions [26], and wildfires [27,28]. It is often used with other models or measurement techniques to provide a comprehensive picture of the air quality in a given area.

The Gaussian plume model was implemented in a parallel Python code using multiprocessing and NumPy modules. The model was based on the Gaussian plume formulation [29], which describes the one-hour average concentration distribution at the surface level as follows:

$$C(x,y) = \frac{E}{2\pi\sigma_y\sigma_z\overline{u}} exp\left(-\frac{y^2}{2\sigma_y^2}\right) exp\left(-\frac{H^2}{2\sigma_z^2}\right) \cdot 1000 \tag{1}$$

where $\overline{u}$ is the one-hour average wind velocity (assumed to be uniform over the whole computational domain), and $\sigma_y(x)$ and $\sigma_z(x)$ are the standard deviations (horizontal and vertical) of the spatial distribution of the plume concentration, which were estimated using dispersion curves for rural areas, as proposed by Briggs [30,31]:

$$\sigma_y(x) = k_{11}x(1 + k_{22}x)^{k_{33}} \tag{2}$$

$$\sigma_z(x) = k_{44}x(1 + k_{55}x)^{k_{66}} \tag{3}$$

The coefficients $k_{11}, \ldots, k_{66}$ depend on the atmospheric stability class (A-F) with a nearly linear growth with a downwind distance $x$ (Table 1).

**Table 1.** Briggs's dispersion curves coefficients for rural areas.

| Stability Class | $k_{11}$ | $k_{22}$ | $k_{33}$ | $k_{44}$ | $k_{55}$ | $k_{66}$ |
|---|---|---|---|---|---|---|
| A | 0.22 | 0.0001 | −0.5 | 0.20 | 0.0 | 0.0 |
| B | 0.16 | 0.0001 | −0.5 | 0.12 | 0.0 | 0.0 |
| C | 0.11 | 0.0001 | −0.5 | 0.08 | 0.0002 | −0.5 |
| D | 0.08 | 0.0001 | −0.5 | 0.06 | 0.0015 | −0.5 |
| E | 0.06 | 0.0001 | −0.5 | 0.03 | 0.0003 | −1.0 |
| F | 0.04 | 0.0001 | −0.5 | 0.016 | 0.003 | −1.0 |

Atmosphere stability was classified based on gradient Richardson number criteria [32]. The gradient Richardson number was estimated based on the meteorological output from the GEM-AQ model. The vertical gradients were calculated between the two lowest layers. Detailed comparison of GEM-AQ meteorological results with the observations can be found in the Appendix A.1.

$H$ is the plume rise above the surface, which is a sum of stack height $H_e$ and plume rise $dh$ calculated using the combination of Holland and CONCAWE formulas ($dh_H$,$dh_C$, respectively) [33], depending on the heat flux $Q$:

$$dh = \begin{cases} dh_C & for\ Q > 24{,}000 \\ dh_H \cdot (24{,}000 - Q)/8000 + dh_C \cdot (Q - 16{,}000)/8000 & for\ Q \in \langle 16{,}000; 24{,}000) \\ dh_H & for\ Q < 16{,}000 \end{cases} \tag{4}$$

### 2.4. Emission Data

The Polish national emission inventory [34] fully covers the study area. Three primary emission SNAP categories from the inventory were used in Gaussian modelling: domestic, industrial and transport emissions. Annual emissions were transformed into monthly emission rates using the weighting factor from annual emission profiles.

Traffic emissions were represented as point sources distributed across 30 m along the road network (Figure 4a). For the uplift formula (Equation (4)), we assumed a fume temperature of 500 K and a velocity of 1 m/s.

Domestic emissions were based on the National Database of Topographic Objects (BDOT), a nationwide system of collecting and sharing topographic data, including vector data describing each building as the basis for the national emission inventory [34]. For the Gaussian model, we assumed a stack height of 3.5 times the number of floors + 0.5 m. The fume temperature was assumed to be 400 K and the velocity was assumed to be 0.5 m/s.

Industrial emissions (Figure 4b) were based on annual reporting obligations provided to the national emission inventory database.

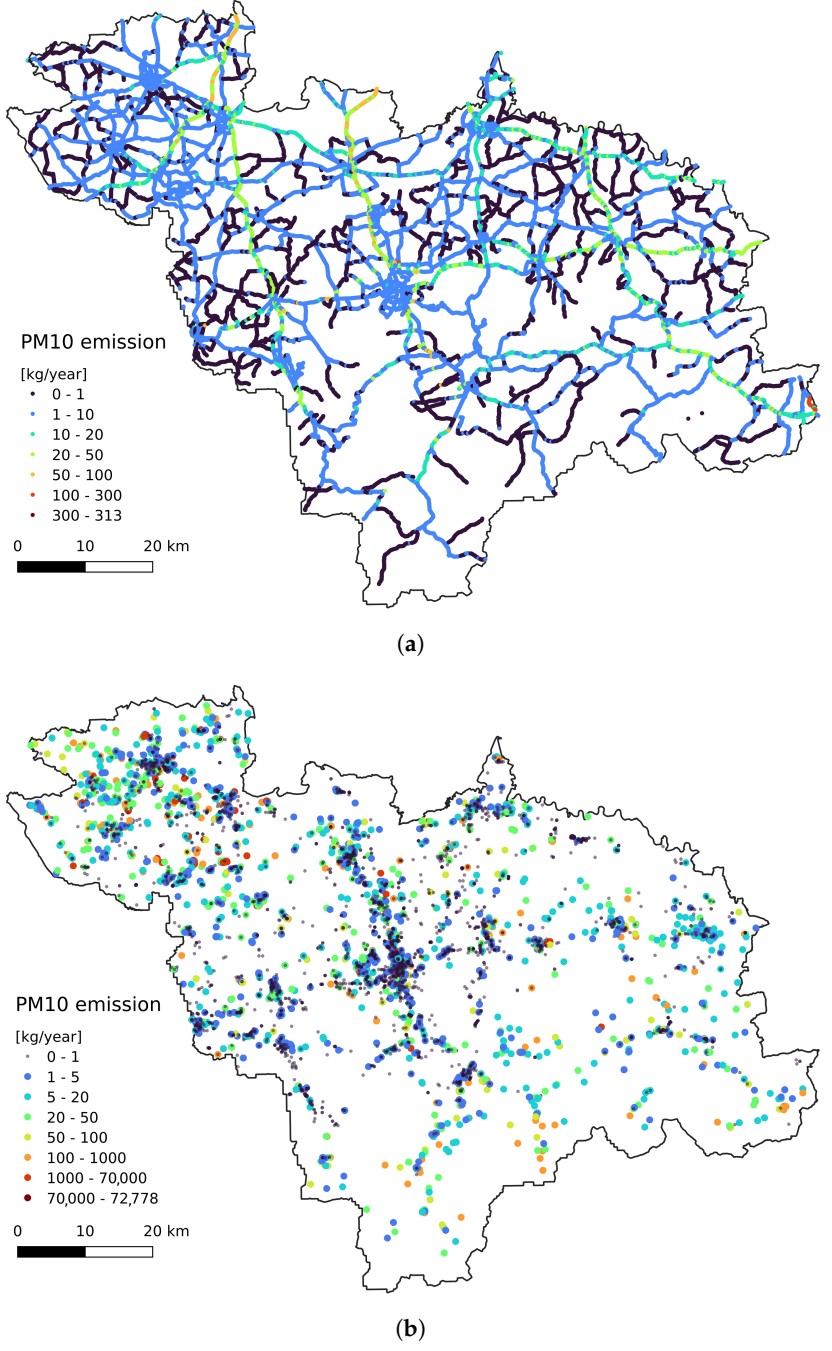

(**a**)

(**b**)

**Figure 4.** *Cont.*

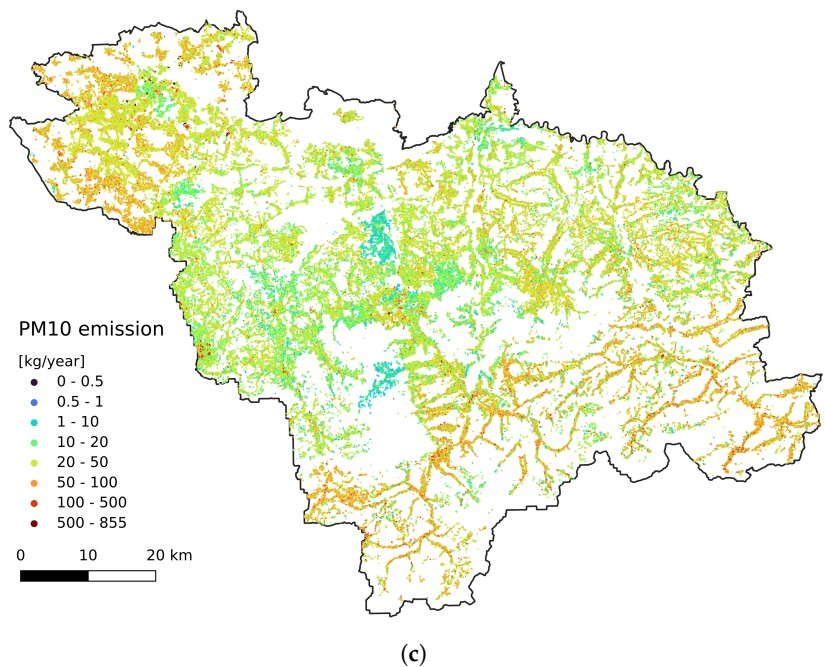

(**c**)

**Figure 4.** $PM_{10}$ annual emissions: (**a**) traffic, (**b**) industrial, (**c**) domestic.

### 2.5. Surface Observations

There are nine air quality stations within the study area. Each measures $PM_{10}$ concentration with an hourly time step. One meteorological station is located in the centre of the area (Figure 1). Table 2 summarizes the annual observed air quality series. Despite the annual mean being at a moderate level (30–40 $\mu g/m^3$), the number of days with the legal threshold (50 $\mu g/m^3$) exceeded is quite significant and covers the central part of the winter season.

**Table 2.** Statistical summary of the observed daily averaged $PM_{10}$ concentration annual time series (2021).

| Observation Station | Mean Concentration | 90.2% Concentration Percentile | No. of Days with a Concentration Exceeding 50 $\mu g/m^3$ |
|---|---|---|---|
| MpOswiecBema | 35.81 | 72.32 | 69 |
| MpSuchaNiesz | 40.9 | 90.6 | 98 |
| SlBielKossak | 29.21 | 52.18 | 48 |
| SlCiesChopin | 31.0 | 56.91 | 54 |
| SlGoczaUzdroMOB | 37.27 | 78.88 | 77 |
| SlRybniBorki | 35.94 | 69.43 | 64 |
| SlUstronSana | 18.03 | 31.62 | 8 |
| SlWodzGalczy | 38.8 | 73.79 | 91 |
| SlZywieKoper | 34.54 | 64.83 | 66 |

As an auxiliary input to random forest training and validation, hourly observations of temperature and wind speed from 2021 were used. Figure 5 summarizes the meteorological observations.

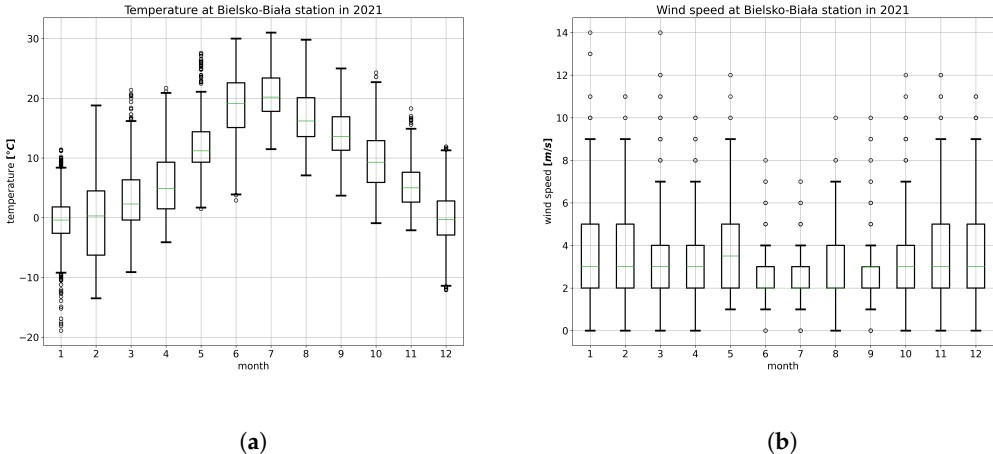

(**a**) 　　　　　　　　　　　　　　　　　　　　　　　(**b**)

**Figure 5.** Observed meteorological conditions at Bielsko Biała: (**a**) temperature; (**b**) wind speed.

### 2.6. Random Forest

As a data fusion algorithm, random forest was used. Random forest is a robust machine learning algorithm for classification [35] and regression tasks [36]. It is a type of ensemble learning algorithm that combines multiple decision trees to improve the predictive performance of the model [37]. Each decision tree in a random forest is constructed using a different subset of the training data and a random subset of the input features. This split introduces diversity and reduces overfitting, as each tree is trained on a different subset of the data and features.

In this work, we trained a random forest algorithm to predict observed $PM_{10}$ concentrations at the observation station locations. Input features included concentrations from GEM-AQ and the Gaussian plume model. The random forest models were trained using a 5-fold cross-validation process. This involves splitting the input data into five subsets, using one subset for evaluation and the remaining four for training, and repeating this five times so that each subset is used once for an evaluation. We also attempted to use calendar-related variables, such as the day of the week and the month, and observed meteorological parameters as additional features. The training dataset was based on time series observations from all nine stations.

As a second trial, we tried using datasets based on observations from a single observation station's time series. This approach is justified because each station's location primarily influences air quality observations, and merging time series from multiple locations may only sometimes be the best approach [38].

Finally, we attempted station vs. station cross-validation, as we anticipated the presence of clusters of similar stations in terms of air quality dynamics within the analyzed area. Additionally, information about outliers (i.e., stations that were different from the others) would help exclude them from the training dataset.

Random forest was implemented using a Python code with the scikit-learn module. The hyperparameters used were 513 estimators (trees), and the tree depth was limited to 5. The absolute error was used as a training criterion.

The trained model was then applied to each pixel of the computational domain (with a 250 m spatial resolution) using the Gaussian plume and GEM-AQ results as input to predict the concentration in a grid cell as output. Depending on the target variable, this process was performed on hourly or daily concentrations (3).

### 3. Results

In the following section we discuss the aggregated measures of various random forest configurations. Detailed comparison of the resulting and observed $PM_{10}$ time series can be found in the Appendix A.3.

### 3.1. Overall Performance

In order to assess the reproducibility of $PM_{10}$ concentration dynamics, we examined the coefficient of determination $R^2$ of the GEM-AQ and Gaussian plume models within a monthly time window (Figures 6 and 7). Both models perform better in winter months (October–March) than in summer (April–September). This pattern is observed at all air quality stations and can be explained by the meteorological factors driving $PM_{10}$ concentration, which is better reproduced in winter than in summer. Another reason might be the temporal emission profile, which is more uniform in winter (due to constant demand for heating) than in summer when the daily profile is not uniform.

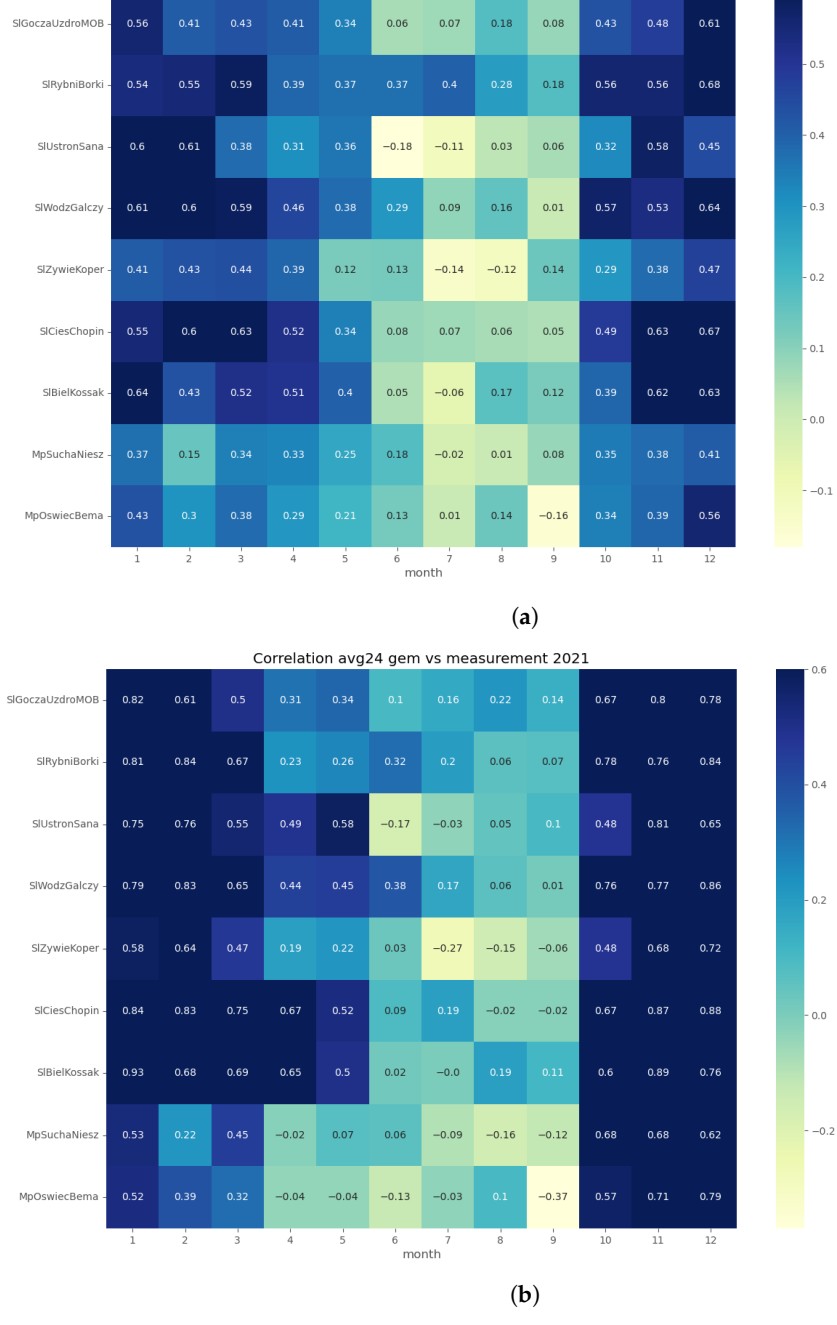

(**a**)

(**b**)

**Figure 6.** R-squared coefficients between observations and GEM-AQ results: (**a**) hourly concentrations; (**b**) daily averaged concentrations.

Both GEM-AQ and Gaussian plume models perform better with daily averaged concentrations. This fact is due to rapid changes in observed concentrations, which cannot be simulated by any of these models [39]. Some authors [3] explain this concentration variation by emission, which is driven by the air temperature.

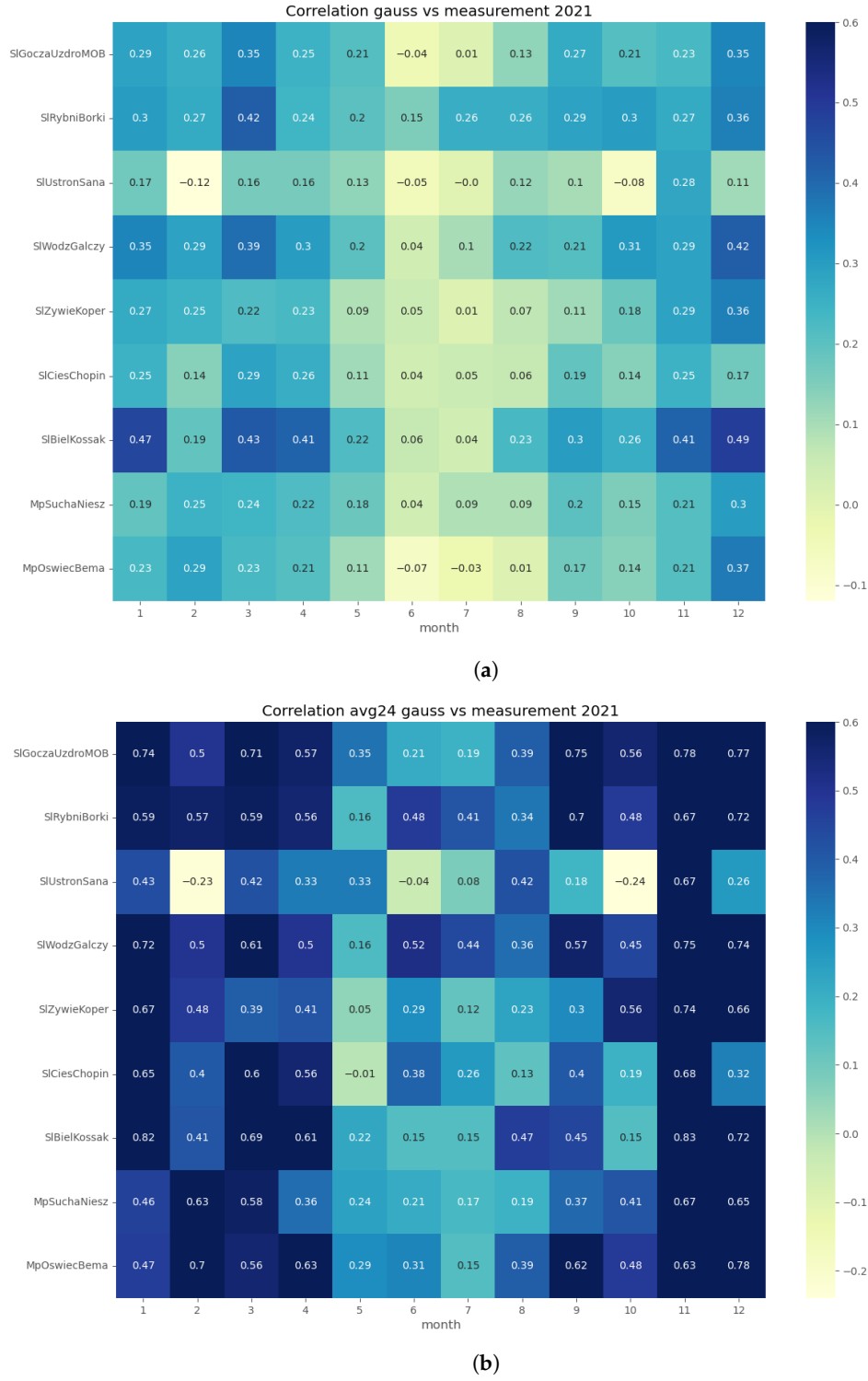

**Figure 7.** R-squared coefficients between observations and Gaussian plume model results: (**a**) hourly concentrations; (**b**) daily averaged concentrations.

The random forest algorithm's performance was evaluated based on $R^2$, which is suitable for assessing the dynamics' reproducibility. For assessing accuracy, the accuracy coefficient ($ACC$) was used, which describes how accurate the results predicted by the random forest ($\hat{y}$) are in comparison to observations ($y$):

$$ACC = \left(1 - \sum \frac{|\hat{y} - y|}{y}\right) \cdot 100\% \tag{5}$$

As Table 3 reveals, reproducing the hourly dynamics of $PM_{10}$ concentrations was challenging, regardless of the extra features. Using daily averaged concentrations instead of hourly concentration increased the $R^2$ from around 0.2 to 0.4 and accuracy from 48% to 60%. Using additional features improved the model's dynamics in all cases, while the accuracy remains almost the same.

**Table 3.** Performance of the initial run of random forest, depending on data aggregation (rows) and additional input features.

| Target Variable | | No Additional Features | Day of the Week, Month | Day of the Week, Month, Observed Wind, Observed Temperature |
|---|---|---|---|---|
| hourly concentration | $R^2$ | 0.28 | 0.34 | 0.37 |
| | $ACC[\%]$ | 44.9 | 48.6 | 48.7 |
| daily mean concentration | $R^2$ | 0.49 | 0.54 | 0.61 |
| | $ACC[\%]$ | 62.8 | 65.5 | 68.1 |
| daily median concentration | $R^2$ | 0.43 | 0.46 | 0.55 |
| | $ACC[\%]$ | 59.9 | 62.6 | 66.0 |
| daily maximum concentration | $R^2$ | 0.43 | 0.45 | 0.48 |
| | $ACC[\%]$ | 57.8 | 59.7 | 60.3 |

*3.2. Temporal Comparison*

We analyzed the performance of random forest models trained on data from one month only. We used 5-fold cross-validation and data from all the observation stations. The process was repeated for hourly data and daily averages. As the data from Table 4 reveal, the best performance in terms of dynamics reproduction ($R^2$) was obtained for winter months. At the same time, the accuracy (and thus, the reproduction of the magnitude of observed concentrations) was better in the summer months. This fact can be explained by a general tendency of ensemble methods, which are not very good at reproducing peak values. Also, some authors claim that a significant amount of emissions is not included in the national emission inventory [40,41].

**Table 4.** Performance of the random forest with data of single months used for training.

| Hourly Concentration | $ACC[\%]$ | $R^2$ | Daily Mean | $ACC[\%]$ | $R^2$ |
|---|---|---|---|---|---|
| January | 45 | 0.21 | January | 61 | 0.36 |
| February | 40 | 0.15 | February | 53 | 0.25 |
| March | 46 | 0.17 | March | 59 | 0.17 |
| April | 52 | 0.14 | April | 70 | 0.06 |

**Table 4.** *Cont.*

| Hourly Concentration | ACC[%] | $R^2$ | Daily Mean | ACC[%] | $R^2$ |
|---|---|---|---|---|---|
| May | 55 | 0.04 | May | 72 | 0.2 |
| June | 64 | 0.02 | June | 78 | 0.02 |
| July | 57 | 0 | July | 71 | −0.07 |
| August | 50 | 0.02 | August | 70 | 0.05 |
| September | 51 | 0.04 | September | 68 | −0.04 |
| October | 44 | 0.13 | October | 60 | 0.14 |
| November | 48 | 0.23 | November | 65 | 0.44 |
| December | 39 | 0.31 | December | 57 | 0.49 |

*3.3. Spatial Comparison*

Finally, we analyzed if the choice of observation station location influenced the performance of the random forest. We used a 5-fold cross-validation process and data from one station at a time. The results make it possible to distinguish stations with sufficiently better performance (SlBielKossak and SlWodzGalczy) and stations with significantly worse performances (MpSuchaNiesz and MpOswiecBema), as can be seen in Table 5. This difference can be explained when we look at the station location. The former is located in dense urban areas with local district heating systems. In contrast, the latter is in a single-family housing area with typical low-stack residential heating emissions.

**Table 5.** Performance of the random forest with single station used for training process.

| Hourly Concentration | ACC[%] | $R^2$ | Daily Mean | ACC[%] | $R^2$ |
|---|---|---|---|---|---|
| SlBielKossak | 60 | 0.5 | | 72 | 0.64 |
| SlWodzGalczy | 56 | 0.46 | | 71 | 0.63 |
| SlRybniBorki | 51 | 0.39 | | 66 | 0.51 |
| SlCiesChopin | 48 | 0.43 | | 69 | 0.72 |
| SlUstronSana | 49 | 0.35 | | 66 | 0.51 |
| SlGoczaUzdroMOB | 50 | 0.36 | | 62 | 0.53 |
| SlZywieKoper | 37 | 0.41 | | 62 | 0.54 |
| MpSuchaNiesz | 34 | 0.37 | | 59 | 0.59 |
| MpOswiecBema | 44 | 0.3 | | 58 | 0.39 |

*3.4. Annual Statistics*

The annual statistics of the GEM-AQ model (Figures 8a, 9a and 10a) resemble the emission pattern (Figure 4). High concentrations are observed in the western part of the study area (Rybnik and Wodzisław cities) and the centre (Bielsko-Biała and Żywiec). The results from the Gaussian plume model highly underestimate the average $PM_{10}$ concentration (Figure 8b). The Gaussian plume model and random forest reveal a complex concentration pattern in the southern part of the study area, resulting from complex topography. The order of the magnitude of the random forest results is similar to that of the GEM-AQ model. The random forest is not good at reproducing peak concentrations; thus, the percentile for the random forest (90.2%) is generally lower than for GEM-AQ (Figure 9). Also, the number of days when the legal threshold of 50 µg/m$^3$ is exceeded is lower for the random forest algorithm than for GEM-AQ.

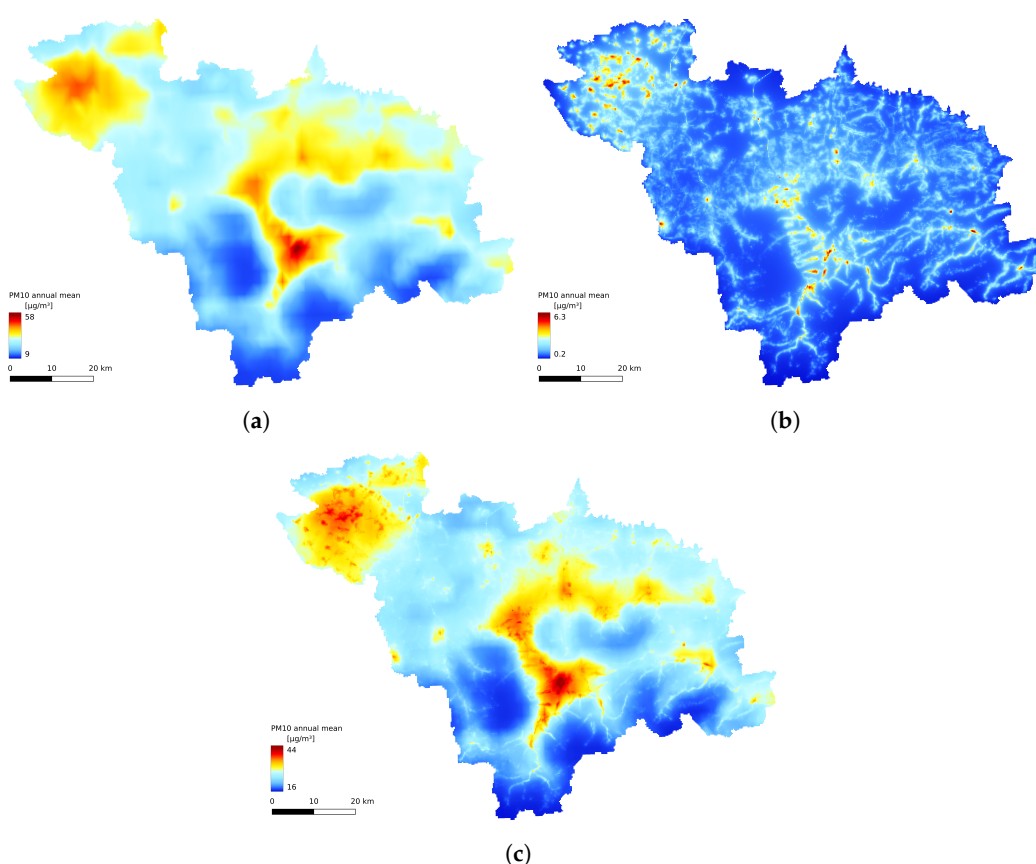

**Figure 8.** Annual average $PM_{10}$ concentration: (**a**) obtained by GEM-AQ model, (**b**) obtained by Gaussian plume model, and (**c**) obtained by random forest.

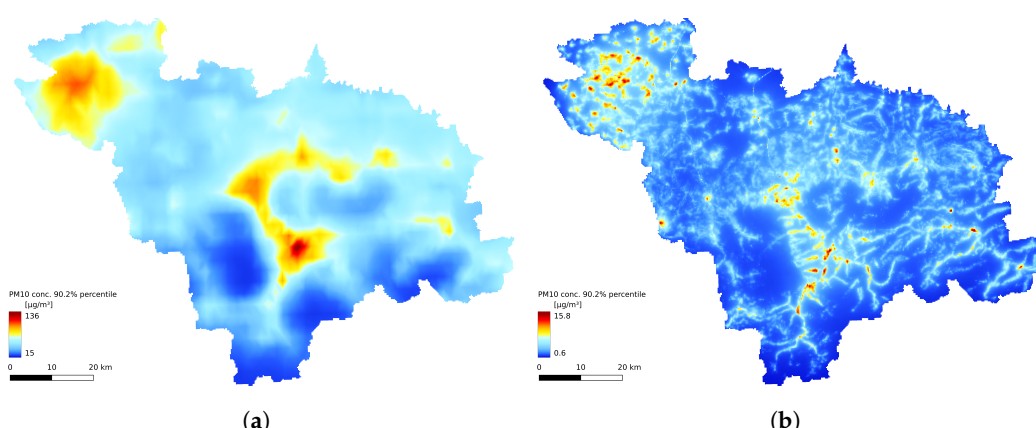

**Figure 9.** *Cont.*

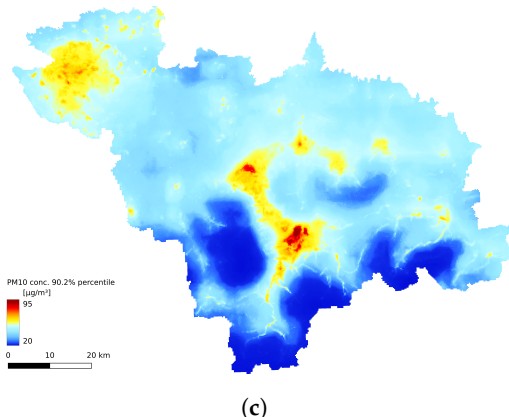

(**c**)

**Figure 9.** The 90.2% percentile of daily averaged $PM_{10}$ concentration timeseries: (**a**) obtained by GEM-AQ results, (**b**) obtained by Gaussian plume model, and (**c**) obtained by random forest model.

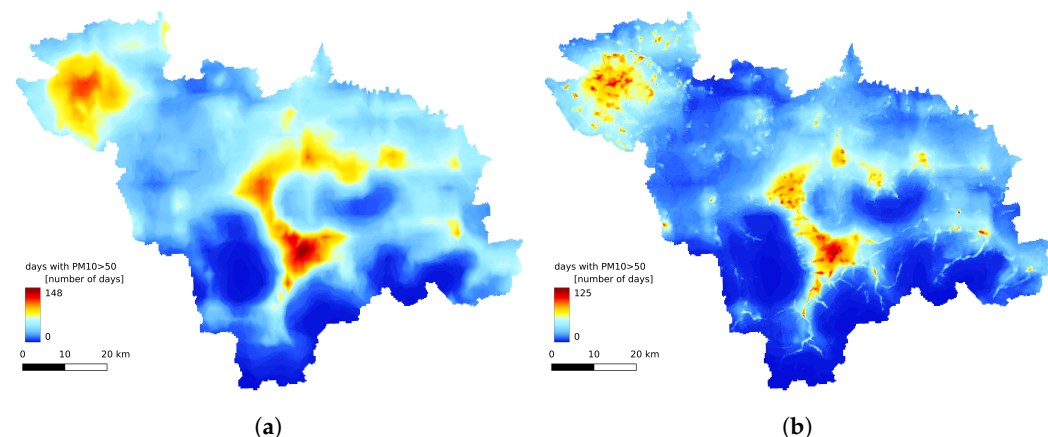

(**a**)                                        (**b**)

**Figure 10.** Number of days with average daily concentration exceeding 50 μg/m³: (**a**) obtained from GEM-AQ model; (**b**) obtained by random forest results.

## 4. Discussion

We have analyzed several variants of a downscaling approach based on random forests. The poorest performance was obtained by an hourly forecasting model with only the GEM-AQ and Gaussian plume models as input. Using calendar variables (day of the week and month) as auxiliary inputs improved the $R^2$ of the model's results. This was due to the seasonal variation of emissions, which was not included in the annual EMEP temporal profile. However, it was *trained* by the random forest based on observations. This includes holiday seasons, heating days (which influence domestic emissions), and work schedules, which determine traffic emissions.

Using meteorological observations as auxiliary data also improved the random forest results. They reduced the uncertainty that was introduced by the meteorological part of GEM-AQ. By knowing the temperature and wind data, the random forest can mitigate some uncertainty from the meteorological model.

Finally, we tried using daily aggregation (mean or median). This approach led us to better results (accuracy and dynamics). Such a result is due to the nature of random forest regression, which tends to generalize training sets. Thus, it is not good at reproducing extreme values (outliers from the statistical point of view). Nevertheless, aggregated values (annual statistics) assess human exposure to pollutants. The lack of reliable hourly results should be acceptable for this application.

Poor results in reproducing hourly dynamics of $PM_{10}$ concentrations may be explained by a combination of several causes, including the spatial representativeness of meteorologi-

cal results and the intrinsic uncertainty of the emission inventory, which, especially in the case of traffic or domestic emissions, is based on approximated emission factors.

## 5. Conclusions

Random forest regression is a powerful and robust technique for developing non-linear regression models. As we have shown, it can be applied to obtain high-resolution concentration maps based on regional model results. As a random forest cannot extrapolate data, its results are slightly underestimated.

The accuracy of random forest improves when applied to daily averaged values. This is likely due to the smoothing effect of averaging, thanks to which no sharp gradients must be simulated. Additional improvements to the random forest regression model can be made by using additional features. Including the day of the week and month improved both accuracy and dynamics of all random forest variants. These features act as a non-explicit temporal profile, which helps to adjust the regression to temporal changes in emissions. Including meteorological observations (temperature and wind) as additional features is also helpful in improving the random forest model's regression results. However, the improvement is less significant in this case. The effect of meteorological observations is likely a way of fixing the inaccuracy of the meteorological results of the GEM-AQ model, which later on affected the air quality results.

The choice of observation stations for random forest training should be made with care. Some stations deliver observations that could be more challenging to replicate. On the other hand, using a single observation station's time series as a training target over large areas may produce results that need more universality.

**Author Contributions:** Conceptualization: J.S.; methodology, software, validation, resources, and data curation: M.K.; writing—original draft preparation: M.K.; writing—review and editing: J.S.; visualization: M.K.; supervision: J.W.K.; project administration, J.W.K. All authors have read and agreed to the published version of the manuscript.

**Funding:** This research received no external funding.

**Institutional Review Board Statement:** Not applicable.

**Informed Consent Statement:** Not applicable.

**Data Availability Statement:** The data utilized in this research article will be made available upon publication of the complete research study. The authors intends to share the data, including any associated supplementary materials, to facilitate transparency, reproducibility, and further scientific inquiry. Furthermore, the authors plans to publish the code related to this research on GitHub as soon as it reaches a level of maturity that ensures its usefulness and comprehensibility to the scientific community. The code repository on GitHub will provide detailed instructions and documentation on how to use the code to replicate and build upon the findings reported in this paper. The availability of both data and code will enable researchers to validate the results, conduct further analysis, and potentially collaborate on related investigations. The authors recognize the importance of open data and open-source software in promoting scientific progress and encourages the utilization of these resources to advance knowledge in the field. Once the data and code are accessible, information on how to access them will be included in the published paper. This will ensure that interested researchers have the necessary means to explore, replicate, and expand upon the findings of this research.

**Conflicts of Interest:** The authors declare no conflicts of interest.

## Appendix A. Timeseries Evaluation

### Appendix A.1. Meteorological Model Evaluation

The meteorological part of the GEM-AQ model is a source of meteorological variables (wind speed, wind direction, gradient Richardson number) for the Gaussian plume model. Within the modelling domain (Figure 1), one meteorological observation station exists (Bielsko-Biała). A comparison of observed and modelled hourly time series are presented in Figure A1. The temporal temperature pattern is reproduced reasonably well. Some

underestimations in temperature are noticeable in winter extremes. The model tends to underestimate the observed wind speed for wind time series. Also, calm conditions with no wind are not reproduced by GEM-AQ.

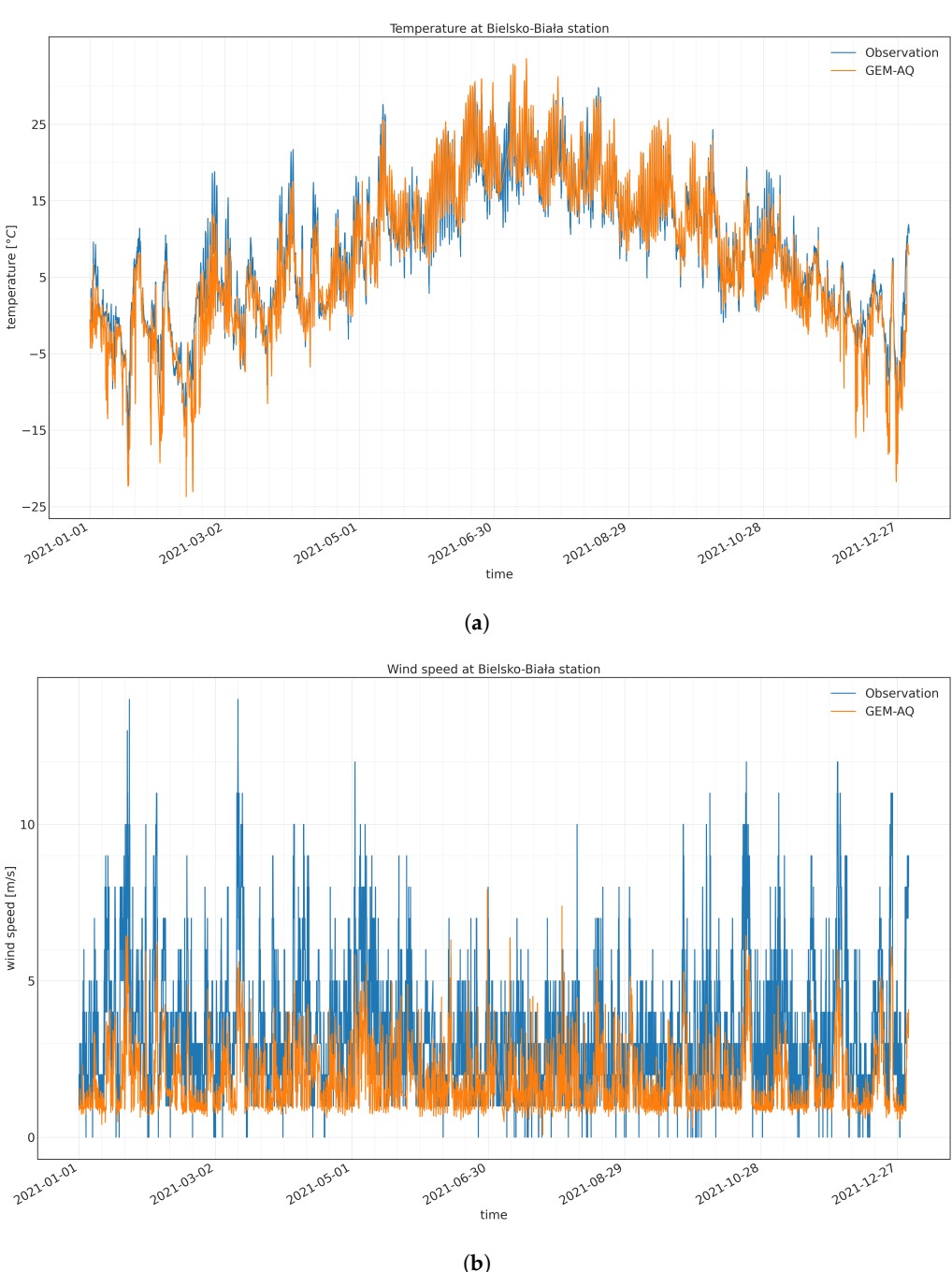

(**a**)

(**b**)

**Figure A1.** Hourly time series of meteorological parameters bserved and modelled by GEM-AQ: (**a**) temperature; (**b**) wind speed.

*Appendix A.2. PM10 Input Time Series Evaluation*

The GEM-AQ and the Gaussian plume model are the sources of the input time series for the random forest algorithm. The observation data from nine observation stations are the target variables for random forest training (with a 5-fold cross-validation procedure). Figures A2–A4 present time series comparisons at each air quality station location.

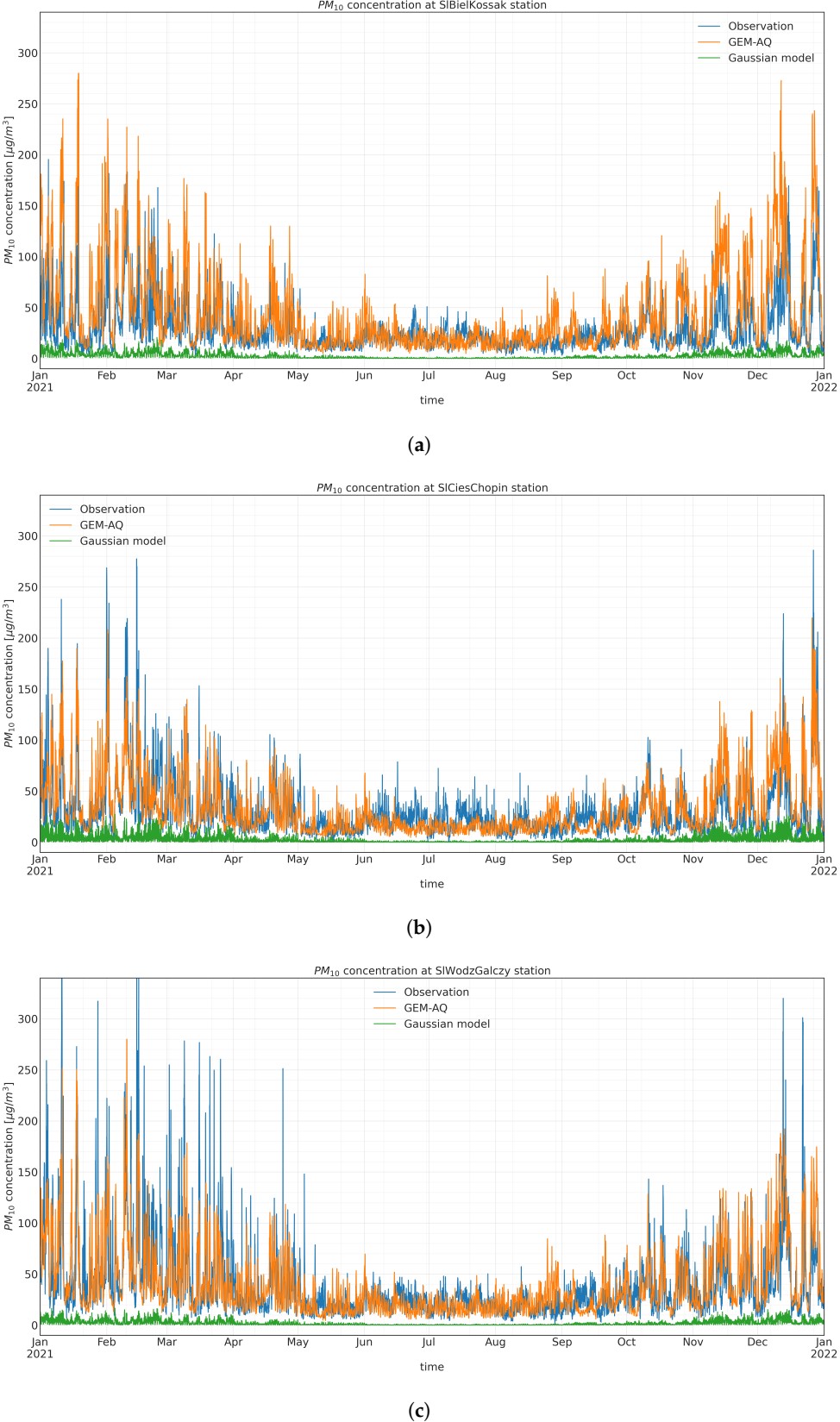

**Figure A2.** Observed and modelled $PM_{10}$ hourly concentration time series at (**a**) SlBielKossak, (**b**) SlCiesChopin, and (**c**) SlWodzGalczy.

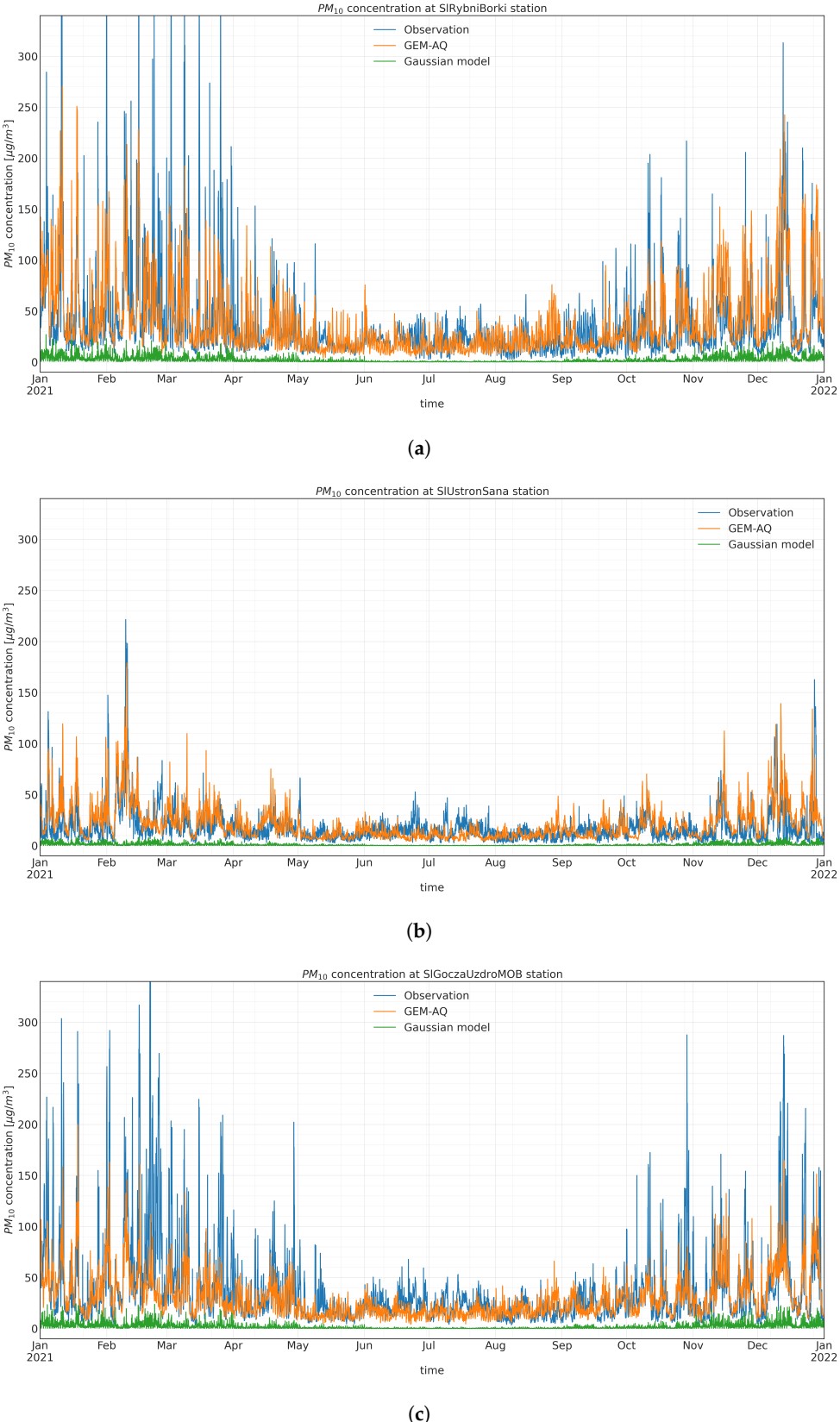

**Figure A3.** Observed and modelled $PM_{10}$ hourly concentration time series at (**a**) SlRybniBorki, (**b**) SlUstronSana, and (**c**) SlGoczaUzdroMOB.

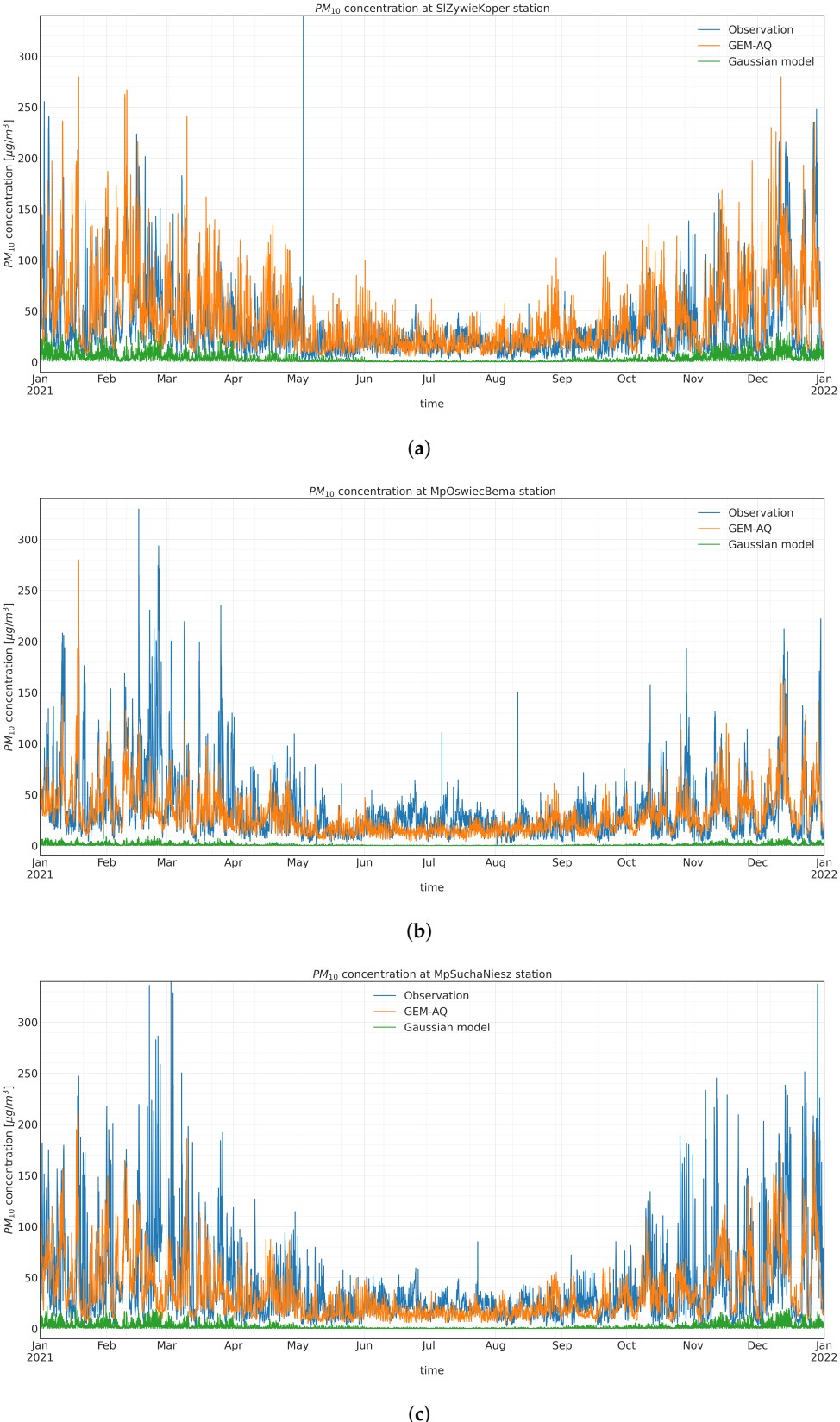

**Figure A4.** Observed and modelled $PM_{10}$ hourly concentration time series at (**a**) SlZywieKoper, (**b**) MpOswiecBema, and (**c**) MpSuchaNiesz.

*Appendix A.3. Random Forest Output $PM_{10}$ Time Series*

Figures A5–A7 present the daily $PM_{10}$ output time series from the most accurate random forest model trained within this work (using daily mean concentrations and auxiliary variables, including the day of the week, month, and meteorological observations; see Table 3).

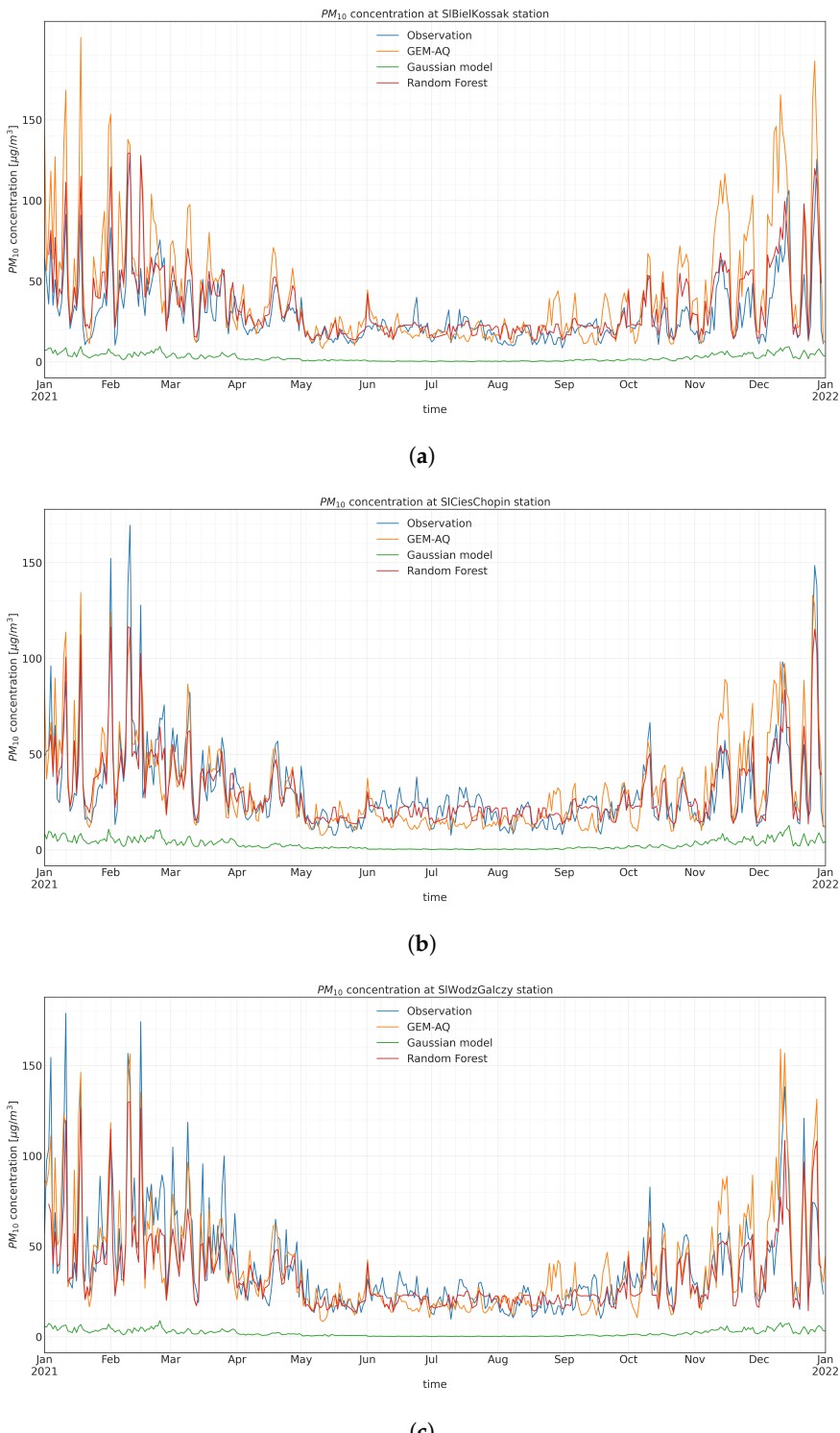

**Figure A5.** Observed and modelled $PM_{10}$ daily concentration time series at (**a**) SlBielKossak, (**b**) Sl-CiesChopin, and (**c**) SlWodzGalczy.

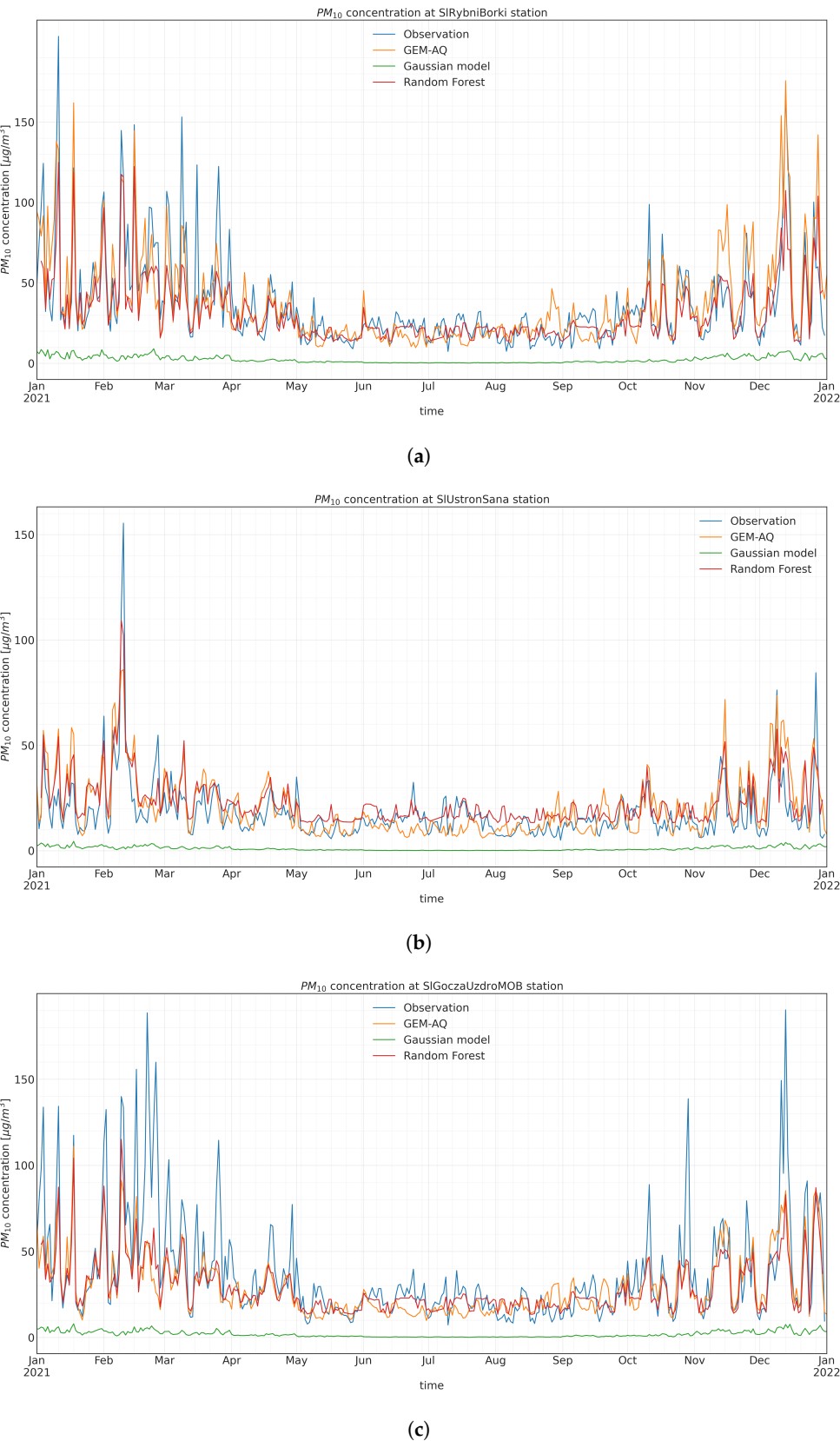

**Figure A6.** Observed and modelled $PM_{10}$ daily concentration time series at (**a**) SlRybniBorki, (**b**) SlUstronSana, and (**c**) SlGoczaUzdroMOB.

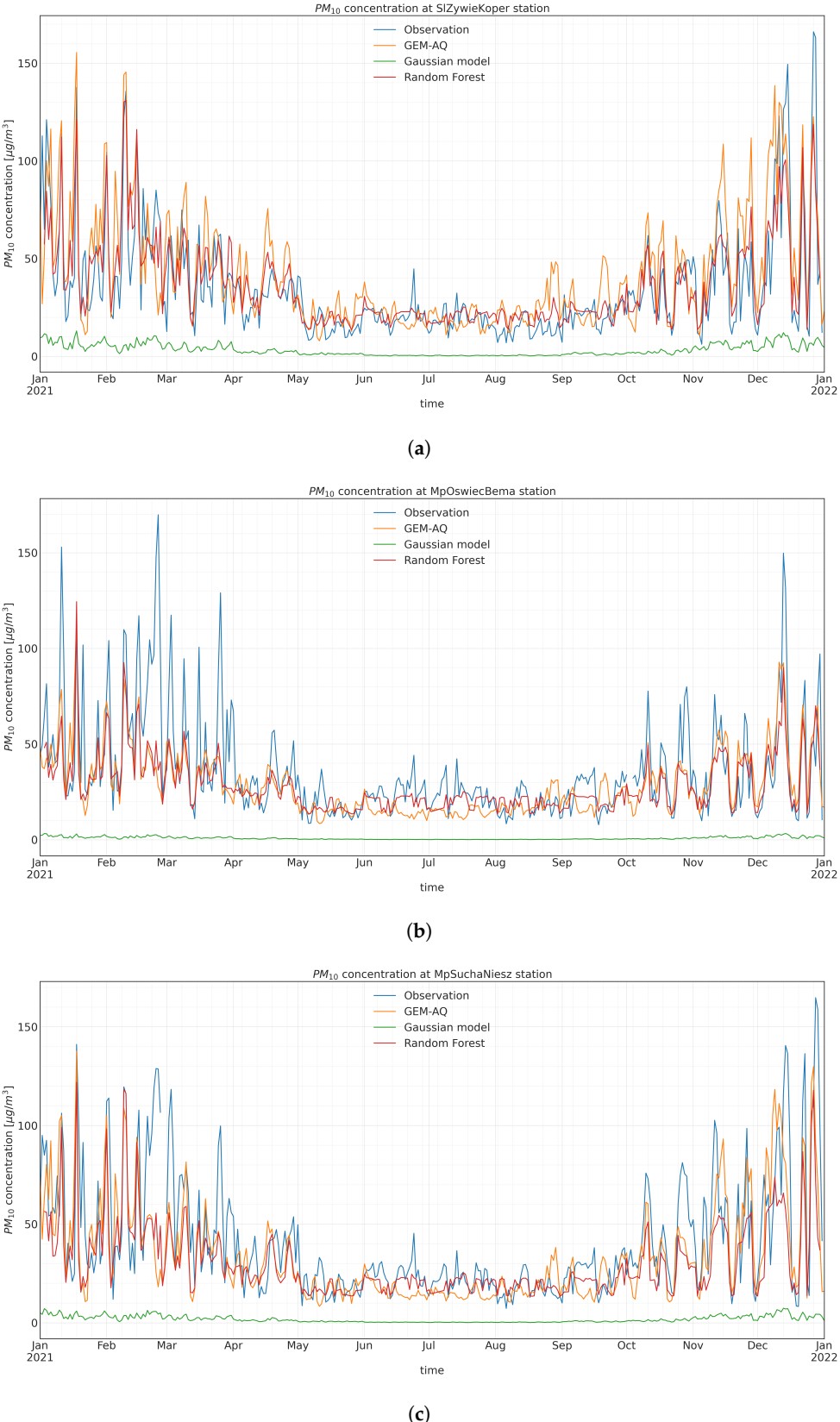

**Figure A7.** Observed and modelled $PM_{10}$ daily concentration time series at (**a**) SlZywieKoper, (**b**) MpOswiecBema, and (**c**) MpSuchaNiesz.

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
