# Peer review of "Downscaling of Regional Air Quality Model Using Gaussian Plume Model and Random Forest Regression"

_atmosphere, doi:10.3390/atmos14071171_

Round 1

Reviewer 1 Report

This paper presents an application of random forests for downscaling regional model air quality results across Poland. The current approach has greater potential to tackle localized impacts of human interventions on PM10 release with greater spatial resolution. Results are presented very comprehensively. Manuscripts can be accepted for publication with minimum effort. 

Comments:

Introduction; Authors need to incorporate more details on the background and significance of the study particularly discussing how the assessment of PM2.5 is important for Poland. How human interventions and industrialization influence the air quality in Poland.

Objectives are not clear...

Figure 2. Label a and b on sub-figures

The downscaling steps are missing. Authors need to clarify them in methods.

Figure 4. Improve resolution. It's unclear to read.

Table 2. ACC??? Describe performance indicators in the method

Improve the resolution of legends for all maps

The results presented are unclear and not sure for which year they are. Also, applications of the framework seem too limited. Why authors have not provided detailed maps of downscaled results from January to December and for some selected days of the year?

Authors can include a discussion section. Describe the main findings, and compare them with the literature. Also, list the shortcomings of the current framework that future studies can consider for further advancement.  

Minor editing of English language required

Author Response

Dear Reviewer

 We appreciate the time and effort you and the reviewers dedicated to providing feedback on our manuscript and are grateful for the insightful comments and valuable improvements to our paper. 

Please find attached PDF file with detailed reply to all your questions/remarks.

Thank you for your constructive remarks.

Reviewer 2 Report

The authors examine and assess the use of Random Forest Regression and Gaussian Plume Models to estimate PM10 concentrations. The paper is well-written but needs a thorough check for English language issues.

Line 1: There is no need to italicize "PM10"

Line 2: No need to italicize "ug/m3"

Line 12: Downscaling is misspelled

Line 60: What is "while year"?

Line 68: "Voivodship" is not very well-known; consider using "province".

Line 69: "mln" is not very widely used - consider "M" or "million".

Figure 1: The legend text seems to be low resolution and is grainy.

Line 91: What does "SNAP" mean?

Figure 2.B: Make the legend numbers a bit larger for reading ease.

Line 121: Is there a reference for the national emission inventory?

Line 124: Is there a reference for the weighting factor?

Line 127: "m/s" does not need to be italicized.

Fig 3.B is not referenced in the main text.

There seems to be a need to Fig 3.C to show Domestic emissions.

Table 1: Correct the last row header and use the "mu" symbol instead of "u".

Line 153: What is "day of the week number"?

Figures 4 and 5: The text is too small - both the axes and the numbers inside the squares. The latter can be increased a few sizes for ease of reading.

Table 2: What does "*" mean next to hourly concentration?

Table 3: "with" is misspelled.

Lines 207-8: The figure references are out of order.

The manuscript need a complete English language check.

Author Response

Dear Reviewer,

 We appreciate the time and effort that you and the reviewers dedicated to providing feedback on our manuscript and are grateful for the insightful comments on and valuable improvements to our paper. All your remarks have been included in the revised version of the manuscript.

Please find enclosed detailed comments to all your remarks.

Thank you once again.

Reviewer 3 Report

General comments:

The manuscript “Downscaling of Regional Air Quality Model using Gaussian Plume Model and Random Forest Regression” by Kawka et al. analyzes the spatio-temporal distribution of PM10 concentration in a region of Poland by applying a Gaussian model and random forest regression. The topic is interesting and extensively studied recently. In this paper, the introduction lacks essential definitions and references to recent studies on the topic are missing. The methods and results also lack fundamental aspects and the discussion of the results is barely hinted at. Also, authors should pay more attention to the submission, as some figures are not visible in the pdf version of the manuscript. Furthermore, the paper seems to be a mere application of downscaling and it is not clear what the innovative element of the paper is.

For the aforementioned reasons, in this version the paper is not considered suitable for publication.

Specific comments:

Introduction:

- The definition of PM and PM10, which is the focus of the paper, is missing

- When numerical dispersion models are introduced, the description is generic and superficial and there is a lack of references regarding recent studies with applications aimed at studying PM10 concentration.

- What is the regional GEM-AQ model? More detailed information is needed in the introduction.

Data and methods

- In Figure 1, the references of Lat. and Lon. of the studied area need to be added.

- Under which meteorological conditions does PM10 accumulate in the studied region?

- As mentioned by the authors in Lines 101-104, the Gaussian plume model is widely used for the study of air quality. It is necessary to add references regarding studies on the topic.

- In the equations it is necessary to introduce all the variables and parameters.

- What values do the coefficients k11..k44 in equations 2 and 3 assume?

- What is the estimation of the temperature and velocity values of the fumes based on?

- In Table 1, it would be helpful to add classification of weather station.

- What is the spatial resolution of the model results?

Results:

- The plots are difficult to read, the labels, ticks and axes must be enlarged as they are illegible.

- Table 3: the discussion of the average concentration alone is not sufficient

- Figures 8-9-10 are not visible in the file.

Author Response

Dear Reviewer

 We appreciate the time and effort you and the reviewers dedicated to providing feedback on our manuscript and are grateful for the insightful comments and valuable improvements to our paper. Thank you for your constructive remarks.

Please find attached PDFfile with detailed replies to your comments.

Kind regards,

Authors 

Reviewer 4 Report

Review of “Downscaling of Regional Air Quality Model using Gaussian Plume Model and Random Forest Regression” by Kawka et al.

Unfortunately, this manuscript by Kawka et al. does not meet minimum standard for research article, in terms of scientific goal, literature review, methodology descriptions, and scientific findings. None of them are appropriately addressed. I recommend that this manuscript should be rejected.

Author Response

Dear Reviewer,

 We appreciate the time and effort you and the reviewers dedicated to providing feedback on our manuscript and are grateful for the insightful comments and valuable improvements to our paper. Nevertheless we did our best to improve the paper. We hope you will find it publishable.

All the best,

Authors.

Author Response

Dear Reviewer,

We appreciate your time and effort to read and review our manuscript. We have included all your remarks in a revised version of the manuscript. We also address you general comments in the attached PDF file. We hope you will find our answers relevant and informative.

Kind regards,

Authors.

Round 2

Reviewer 3 Report

The authors followed the suggestions of the Reviewer and, in this form, the paper is publishable

Author Response

Thank you for your review.

Reviewer 4 Report

Review of “Downscaling of Regional Air Quality Model using Gaussian Plume Model and Random Forest Regression” by Kawka et al.

I appreciate the authors' efforts to revise the manuscript. However, after reviewing the revised version, I still cannot recommend the manuscript for publication.

The reconstruction of a chemical field over a region necessitates a meticulously designed study and data processing procedures. This is because such a study should provide fundamental information for research into public health and environmental impacts. As it stands, I wouldn't consider using the results of this manuscript because I don't believe the study has been properly validated. While I don't claim the manuscript to be entirely without value, I see no evidence that suggests its usefulness.

For a manuscript to constitute a complete research article, it needs a comprehensive redesign, addressing at least several key points.

1. Given that the manuscript heavily depends on the use of two modeling systems for training, the authors should provide detailed information on the general performance of these models. Merely displaying monthly correlation coefficients does not sufficiently assure that the models can realistically reproduce the chemical environment of the region. The authors should present hourly time series of model PM and pollutant concentrations, comparing them with actual observations. The performance of the meteorological model should be validated in the same way.

2. I could not find detailed information on the application of the random forest machine learning technique. I request that the authors explicitly state the variables used in the training, and discuss the impact of including such variables. Given the multitude of research currently employing machine learning techniques, what novel approaches can the authors claim for this study?

3. The evaluation of the reconstructed concentrations must be adequately demonstrated. The crux of the matter is that the results should replicate real-world concentrations, both spatially and temporally.

4. The manuscript includes several unnecessary contents. Numerous descriptions of the Gaussian model, especially in table 1 and equations 2-4, seem inappropriate. Instead, the authors should provide the performance of the Gaussian model. Figure 5 also appears unnecessary without actual observations - I am not interested in observing the seasonal variation of temperature and wind speed. I would prefer to see how the meteorological model performs in different seasons.

Author Response

see attached pdf

Reviewer 5 Report

No comments

Author Response

see attached pdf
